# Credit Assignment For Collective Multiagent RL With Global Rewards

**Duc Thien Nguyen**    **Akshat Kumar**    **Hoong Chuin Lau**
School of Information Systems
Singapore Management University
80 Stamford Road, Singapore 178902
{dtnguyen.2014,akshatkumar,hclau}@smu.edu.sg

## Abstract

Scaling decision theoretic planning to large multiagent systems is challenging due to uncertainty and partial observability in the environment. We focus on a multiagent planning model subclass, relevant to urban settings, where agent interactions are dependent on their "collective influence" on each other, rather than their identities. Unlike previous work, we address a general setting where system reward is *not decomposable* among agents. We develop collective actor-critic RL approaches for this setting, and address the problem of multiagent credit assignment, and computing low variance policy gradient estimates that result in faster convergence to high quality solutions. We also develop *difference rewards* based credit assignment methods for the collective setting. Empirically our new approaches provide significantly better solutions than previous methods in the presence of global rewards on two real world problems modeling taxi fleet optimization and multiagent patrolling, and a synthetic grid navigation domain.

## 1   Introduction

Sequential multiagent decision making allows multiple agents operating in an uncertain, partially observable environment to take coordinated decision towards a long term goal [15]. Decentralized partially observable MDPs (Dec-POMDPs) have emerged as a rich framework for cooperative multiagent planning [8], and are applicable to several domains such as multiagent robotics [4], multiagent patrolling [19] and vehicle fleet optimization [42]. Scalability remains challenging due to NEXP-Hard complexity even for two agent systems [8]. To address the complexity, various models are explored where agent interactions are limited by design by enforcing various conditional and contextual independencies such as transition and observation independence among agents [7, 24] where agents are coupled primarily via joint-rewards, event driven interactions [6], and weakly coupled agents [34, 44]. However, their impact remains limited due to narrow application scope.

Recent multiagent planning research has focused on models where agent interactions are primarily dependent on agents' "collective influence" on each other rather than their identities [42, 33, 30, 25, 26]. Such models are widely applicable in urban system optimization problems due to the insight that urban systems are often composed of a *large number* of *nearly identical* agents, such as taxis in transportation, and vessels in a maritime traffic setting [2]. In our work, we focus on the collective Dec-POMDP model ($\mathbb{C}$Dec-POMDP) that formalizes such collective multiagent planning [25], and also generalizes "anonymity planning" models [42]. The $\mathbb{C}$Dec-POMDP model is based on the idea of *partial exchangeability* [13, 27], and collective graphical models [31, 35]. Partial exchangeability in probabilistic inference is complementary to the notion of conditional and contextual independence, and combining all of them leads to a larger class of tractable models and inference algorithms [27].

When only access to a domain simulator is available without exact model definition, several multi-agent RL (MARL) approaches are developed such as independent Q-learning [38], counterfactual multiagent policy gradients and actor-critic methods [16, 23], multiagent Q-learning [29], SARSA-based MARL for Dec-POMDPs [14], and MARL with limited communication [47, 48]. However, most of these approaches are limited to tens of agents in contrast to the collective setting with thousands of agents, which is the setting we target. Closely related to the collective setting we address, special MARL sub-classes are proposed to model and control population-level behaviour of agents such as mean field RL (MFRL) [46] and mean field games (MFGs) [45, 17]. MFGs are used learn the behaviour of a population of agents in an inverse RL setting. The MFRL framework does not explicitly address credit assignment, and also requires agents to maintain individual state-action trajectories, which may not be scalable with thousands of agents, as is the case in our tested domains.

We focus on the problem of learning agent policies in a MARL setting for $\mathbb{C}$Dec-POMDPs. We address the crucial challenge of *multiagent credit assignment* in the collective setting when joint actions generate a team reward that *may not* be decomposable among agents. The joint team rewards make it difficult for agent to deduce their individual contribution to the team's success. Such team reward settings have been recognised as particularly challenging in the MARL literature [9, 16], and are common in disaster rescue domains (ambulance dispatch, police patrolling) where the penalty of not attending to a victim is awarded to the whole team, team games such as StarCraft [16], and traffic control [39]. Previous work in $\mathbb{C}$Dec-POMDPs develops an actor-critic RL approach when the joint reward is additively decomposable among agents [25], and is unable to address non-decomposable team rewards. Therefore, we develop multiple actor-critic approaches for the general *team reward* setting where some (or all) joint-reward component may be non-decomposable among agents. We address two crucial issues—multiagent credit assignment, and computing low variance policy gradient estimates for faster convergence to high quality solutions even with thousands of agents. As a baseline approach, we first extend the notion of *difference rewards* [41, 16], which are a popular way to perform credit assignment, to the collective setting. Difference rewards (DRs) provide a conceptual framework for credit assignment; there is no general computational technique to compute DRs in different settings. Naive extension of the previous DR methods in deep multiagent RL setting [16] is infeasible for large domains. Therefore, we develop novel approximation schemes that can compute DRs in the collective case even with thousands of agents.

We show empirically that DRs can result in high variance policy gradient estimates, and are unable to provide high quality solutions when the agent population is small. We therefore develop a new approach called *mean collective actor critic* (MCAC) that works significantly better than DRs and MFRL across a range of agent population sizes from 5 to 8000 agents. The MCAC analytically marginalizes out the actions of agents by using an approximation of the critic. This results in low variance gradient estimates, allows credit assignment at the level of gradients, and empirically performs better than DR-based approaches.

We test our approaches on two real world problems motivated by supply-demand taxi matching problem (with 8000 taxis or agents), and police patrolling for incident response in the city. We use real world data for both these problem for constructing our models. We also test on a synthetic grid navigation domain. Thanks to the techniques for credit assignment and low variance policy gradients, our approches converge to high quality solutions significantly faster than the standard policy gradient method and the previous best approach [26]. For the police patrolling domain, our approach provides better quality than a strong baseline static allocation approach that is computed using a math program [10].

## 2 Collective Decentralized POMDP Model

We describe the $\mathbb{C}$Dec-POMDP model [25] . The model extends the statistical notion of *partial exchangeability* to multiagent planning [13, 27]. Previous works have mostly explored only conditional and contextual independences in multiagent models [24, 44]. $\mathbb{C}$Dec-POMDPs combine both conditional independences and partial exchangeability to solve much larger instances of multiagent decision making.

**Definition 1** ([27]). *Let* $\mathbf{X} = \{X_1, \ldots, X_n\}$ *be a set of random variables, and* $\mathbf{x}$ *denote an assignment to* $\mathbf{X}$. *Let* $\mathcal{D}_i$ *denote the domain of the variable* $X_i$, *and let* $\mathcal{T} : \times_{i=1}^n \mathcal{D}_i \to \mathcal{S}$ *be a statistic of* $\mathbf{X}$, *where* $\mathcal{S}$ *is a finite set. The set of random variables* $\mathbf{X}$ *is partially exchangeable w.r.t. the statistic* $\mathcal{T}$ *if and only if* $\mathcal{T}(\mathbf{x}) = \mathcal{T}(\mathbf{x}')$ *implies* $Pr(\mathbf{x}) = Pr(\mathbf{x}')$.

In the $\mathbb{C}$Dec-POMDP model, agent identities do not matter; different model components are only affected by agent's local state-action, and a statistic of other agents' states-actions. There are $M$ agents in the environment. An agent $m$ can be in one of the states $i \in S$. We also assume a global state component $d \in D$. The joint state space is $\times_{m=1}^{M} S \times D$. The component $d$ typically models variables common to all the agents such as demand in the supply-demand matching case or location of incidents in the emergency response. Let $\boldsymbol{s}_t, \boldsymbol{a}_t$ denote the joint state-action of agents at time $t$. The joint-state transition probability is:

$$P(\boldsymbol{s}_{t+1}, d_{t+1} | \boldsymbol{s}_t, d_t, \boldsymbol{a}_t) = P_g(d_{t+1} | d_t, \mathcal{T}(\boldsymbol{s}_t, \boldsymbol{a}_t)) \prod_{m=1}^{M} P_l(\boldsymbol{s}_{t+1}^m | \boldsymbol{s}_t^m, \boldsymbol{a}_t^m, \mathcal{T}(\boldsymbol{s}_t, \boldsymbol{a}_t), d_t)$$

where $\boldsymbol{s}_t^m, \boldsymbol{a}_t^m$ denote agent $m$'s local state, action components, and $\mathcal{T}$ is a statistic of the corresponding random variables (defined later). We assume that the local state transition function is the same for all the agents. Such an expression conveys that only the statistic $\mathcal{T}$ of the joint state-action, and an agent's local state-action are sufficient to predict the agent's next state.

**Observation function:** We assume a decentralized and partially observable setting in which each agent receives only a *partial* observation about the environment. Let the current joint-state be $(\boldsymbol{s}_t, d_t)$ after the last join-action, then the observation for agent $m$ is given using the function $o_t(\boldsymbol{s}_t^m, d_t, \mathcal{T}(\boldsymbol{s}_t))$. In the taxi supply-demand case, the observation for a taxi in location $z$ can be the local demand in zone $z$, and the counts of other taxis in $z$ and neighbouring zones of $z$. No agent has a complete view of the system.

The reward function is $r(\boldsymbol{s}_t, d_t, \boldsymbol{a}_t) = \sum_m r_l(\boldsymbol{s}_t^m, \boldsymbol{a}_t^m, d_t, \mathcal{T}(\boldsymbol{s}_t, \boldsymbol{a}_t)) + r_g(d_t, \mathcal{T}(\boldsymbol{s}_t, \boldsymbol{a}_t))$ where $r_l$ is the local reward for individual agents, and $r_g$ is the non-decomposable global reward. Given that the reward function $r_l$ is the same for all the agents, we can further simplify it as $\sum_{i,j} n(i,j) r_l(i, j, d_t, \mathcal{T}(\boldsymbol{s}_t, \boldsymbol{a}_t)) + r_g(d_t, \mathcal{T}(\boldsymbol{s}_t, \boldsymbol{a}_t))$, where $n(i,j)$ is the number of agents in state $i$ and taking action $j$ given the joint state-action $(\boldsymbol{s}_t, \boldsymbol{a}_t)$. We assume that the initial state distribution, $b_o(i) \forall i \in S$, is the same for all the agents; initial distribution over global states is $b_o^g(d) \forall d$.

The above defined model components can also differentiate among agents by using the notion of *agent types*, which can be included in an agent's state-space $S$, and each agent can receive its type as part of its observation. In the extreme case, each agent would be of a different type representing a fairly general multiagent planning problem. However, the main benefit of the model lies in settings when agent types are much smaller than the number of agents.

We consider a finite-horizon problem with $H$ time steps. Each agent has a non-stationary reactive policy that takes as input agent's current state $i$ and the observation $o$, and outputs the probability of the next action $j$ as $\pi_t^m(j|i, o)$. Such a policy is analogous to finite-state controllers in POMDPs and Dec-POMDPs [28, 3]. Let $\pi = \langle \pi^1, \ldots, \pi^M \rangle$ denote the joint-policy. The goal is to optimize the value $V(\pi) = \sum_{t=1}^{H} \mathbb{E}[r_t | b_o, b_o^d]$.

**Global rewards:** The key difference from previous works [25, 26] is that in our model we have a global reward signal $r_g$ that is not decomposable among individual agents, which is crucial to model real world applications. Consider a real world multiagent patrolling problem in figure 1. A set of homogeneous police patrol cars (or agents) are stationed in predefined geographic regions to respond to incidents that may arise over a shift (say 7AM to 7PM). When an incident comes, the central command unit dispatches the closet patrol car to the incident location. The dispatched car becomes unavailable for some amount of time (including travel and incident service time). To cover for the engaged car, other available patrol cars from nearby zones may need to reallocate themselves so that no zones are left vulnerable. The reward in

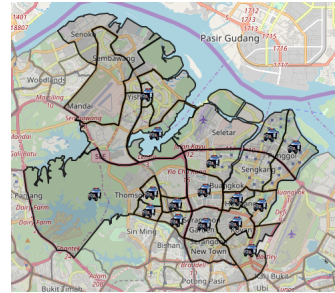

Figure 1: Solid black lines define 24 patrolling zones of a city district

this system depends on the response time to incidents (e.g., threshold to attend to urgent incidents is 10 min, non-urgent in 20 min). The goal is to compute a reallocation policy for agents to minimize the number of unsatisfied incidents where the response time was more than the specified threshold. To model this objective, we award penalty -10 whenever the response time requirement of an incident is not met and 0 otherwise. In this domain, the delay penalty is non-decomposable among patrol cars. It is not reasonable to attribute penalty in an incident to its assigned agent because delay

is due to the intrinsic system-wide supply-demand mismatch. Furthermore, individual agent penalties may even discourage agents to go to nearby critical sectors, which is undesirable (we observed it empirically). Indeed, in this domain, all rewards are global, therefore, previous approaches that require local rewards for agents are not applicable. This is precisely the gap our work targets, and significantly increases the applicability of multiagent decision making to real world applications.

**Statistic for Planning:** We now describe the statistic $\mathcal{T}$ which increases scalability and the generalization ability of solution approaches. For a given joint state-action $(\boldsymbol{s}_t, \boldsymbol{a}_t)$, we define $\mathcal{T}(\boldsymbol{s}_t, \boldsymbol{a}_t) = (n_t(i, j) \forall i \in S, j \in A)$ where each entry $n_t(i, j) = \sum_m I\{(\boldsymbol{s}_t^m, \boldsymbol{a}_t^m) = (i, j)\}$ counts the number of agents that are in state $i$ and take action $j$. We can similarly define $\mathcal{T}(\boldsymbol{s}_t) = (n_t(i) \forall i \in S)$ that counts the number of agents in each state $i$. For clarity, we denote $\mathcal{T}(\boldsymbol{s}_t, \boldsymbol{a}_t)$ or state-action count table as $\mathrm{n}_t^{\mathrm{sa}}$, and the state count table as $\mathrm{n}_t^{\mathrm{s}}$. Given a transition $(\boldsymbol{s}_t, \boldsymbol{a}_t, \boldsymbol{s}_{t+1})$, we define the count table $\mathrm{n}_t^{\mathrm{sas}} = (n_t^{\mathrm{sas}}(i, j, i') \forall i, i' \in S, j \in A)$ which counts the number of agents in state $i$ that took action $j$ and transitioned to next state $i'$. Complete count table is denoted as $\mathbf{n}_t = (\mathrm{n}_t^{\mathrm{s}}, \mathrm{n}_t^{\mathrm{sa}}, \mathrm{n}_t^{\mathrm{sas}})$.

In collective planning settings, the agent population size is typically very large ($\approx 8000$ for our real world experiments). Given such a large population, it is infeasible to compute unique policy for each agent. Therefore, similar to previous works [43, 26], our goal is to compute a homogenous stochastic policy $\pi_t(j|i, o_t(i, d_t, \mathrm{n}_t^s))$ that outputs the probability of each action $j$ given an agent in state $i$ receiving local observation $o_t$ depending on the state variable $d_t$ and state-counts $\mathrm{n}_t^s$ at time $t$. As the policy $\pi$ is dependent on count based observations, it represents an expressive class of policies. Let $\mathbf{n}_{1:H}$ be the combined vector of count tables over all the time steps. Let $\Omega_{1:H}$ be the space of consistent count tables satisfying constraints:

$$\forall t : \sum_{i \in S} \mathrm{n}_t^{\mathrm{s}}(i) = M \ \ ; \sum_{j \in A} \mathrm{n}_t^{\mathrm{sa}}(i, j) = \mathrm{n}_t^{\mathrm{s}}(i) \ \forall i \in S$$
$$\sum_{i' \in S} n_t^{\mathrm{sas}}(i, j, i') = \mathrm{n}_t^{\mathrm{sa}}(i, j) \forall i \in S, j \in A$$
$$\sum_{i \in S, j \in A} \mathrm{n}_t^{\mathrm{sas}}(i, j, i') = n_{t+1}^{\mathrm{s}}(i') \ \forall i' \in S \ ; \tag{1}$$

Count tables $\mathbf{n}_{1:H}$ are the *sufficient statistic* for planning for $\mathbb{C}$Dec-POMDPs.

**Theorem 1.** *[25] The joint-value function of a policy $\pi$ over horizon $H$ given by the expectation of joint reward $r$, $V(\pi) = \sum_{t=1}^{H} \mathbb{E}[r_t]$, can be computed by the expectation over counts:*

$$V(\pi) = \sum_{\mathbf{n}_{1:H} \in \Omega_{1:H}, d_{1:H}} P(\mathbf{n}_{1:H}, d_{1:H}; \pi) \left[ \sum_{t=1}^{H} r_t(\mathrm{n}_t^{\mathrm{sa}}, d_t) \right] \tag{2}$$

*where $r_t(\mathrm{n}_t^{\mathrm{sa}}, d_t) = \sum_{i,j} \mathrm{n}_t^{\mathrm{sa}}(i, j) r_l(i, j, d_t, \mathrm{n}_t^{\mathrm{sa}}) + r_g(d_t, \mathrm{n}_t^{\mathrm{sa}})$*

We show in appendix how to sample from the distribution $P(\mathbf{n}_{1:H}, d_{1:H}; \pi)$ directly *without* sampling individual state-action trajectories.

**Scalability to large agent population:** Since sampling individual agent trajectories is not required, the count-based computation can scale to thousands of agents. Such benefits also extends to computing policy gradients which also depend only on counts $\mathbf{n}$. Furthermore, different data structures and function approximators such as the policy $\pi$ and action-value function depend only on counts $\mathbf{n}$. Such a setting is computationally tractable because if we change only the agent population, the dimensions of the count table still remains fixed, only the counts of agents in different buckets (e.g. $\mathrm{n}(i), \mathrm{n}(i, j)$) changes. Such count-based formulations also extend the generalization ability of RL approaches as multiple joint state-actions $(\boldsymbol{s}_t, \boldsymbol{a}_t, d_t)$ can give rise to the same statistic $\mathbf{n}$. Our goal is to compute the optimal policy $\pi$ that maximizes $V(\pi)$.

**Learning framework:** We follow a *centralized learning* and *decentralized execution* RL framework. Such centralized learning is possible in the presence of domain simulators [16, 23]. We assume access only to a domain simulator that provides count samples $\mathrm{n}$ and the team reward $r$. During centralized training, we have access to all the count-based information, which helps define a centralized action-value function resulting in faster convergence to good solutions. During policy execution, agents execute individual policies without accessing centralized functions . In single agent RL, agent experiences the tuple $(s_t, a_t, s_{t+1}, r_t)$ by interacting with the environment. In the collective case, given that the sufficient statistic is counts, we simulate and learn at the abstraction

of counts. The experience tuple for the centralized learner is $(\mathrm{n}_t^{\mathrm{s}}, d_t, \mathrm{n}_t^{\mathrm{sa}}, \mathrm{n}_t^{\mathrm{sas}}, d_{t+1}, r_t)$. The current joint-state statistic is $(\mathrm{n}_t^{\mathrm{s}}, d_t)$; observations are generated for agents from this statistic and fed into policies. The output is state-action counts $\mathrm{n}_t^{\mathrm{sa}}$. As a result of this joint action, agents transition to new states, and their joint transitions are recorded in the table $\mathrm{n}_t^{\mathrm{sas}}$. Given the constraint set $\Omega$ in (1), the next count table $\mathrm{n}_{t+1}^{\mathrm{s}}$ is computed from $\mathrm{n}_t^{\mathrm{sas}}$; $d_{t+1}$ is the next global state. The joint-reward is $r_t(\mathrm{n}_t^{\mathrm{sa}}, d_t)$. Appendix shows how simulator generates such count-based samples.

**Actor Critic based MARL:** We follow an actor-critic (AC) based policy gradient approach [21]. The policy $\pi$ is parameterized using $\theta$. The parameters are adjusted to maximize the objective $J(\theta) = \mathbb{E}[\sum_{t=1}^{H} r_t]$ by taking steps in the direction of $\nabla_\theta J(\theta)$, which is shown in [26] as:

$$\nabla_\theta J(\theta) = \sum_{t=1}^{H} \mathbb{E}_{d_t, \mathrm{n}_t^{\mathrm{sa}} | b_o, b_o^d, \pi} \left[ Q_t^\pi(\mathrm{n}_t^{\mathrm{sa}}, d_t) \left( \sum_{i \in S, j \in A} \mathrm{n}_t^{\mathrm{sa}}(i,j) \nabla_\theta \log \pi_t(j|i, o(i, d_t, \mathrm{n}_t^{\mathrm{s}})) \right) \right] \quad (3)$$

where $Q_t^\pi$ is the expected return $\mathbb{E}\left[ \sum_{T=t}^{H} r_T | d_t, \mathrm{n}_t^{\mathrm{sa}} \right]$. The above expression can be evaluated by sampling counts n. In the AC approach, the policy $\pi$ is termed as *actor*. We can estimate $Q_t^\pi$ using empirical returns, but it has high variance. To remedy this, AC methods often use a function approximator for $Q^\pi$ (say $Q_w$), which is termed as the *critic*. We consider the critic $Q_w$ to be a continuous function (e.g., a deep neural network) instead of a function defined only for integer inputs. This allows us to compute the derivative of $Q_w$ with respect to all the input variables, which will be useful later. The critic can be learned from empirical returns using temporal-difference learning. We next show several techniques to estimate the collective policy gradient $\nabla_\theta J(\theta)$ that help in the credit assignment problem and provide low variance gradient estimates even for very large number of agents.

# 3   Difference Rewards Based Credit Assignment

Difference rewards provide a powerful way to perform credit assignment when there are several agents, and have been explored extensively in the MARL literature [41, 1, 39, 40, 12]. Difference rewards (DR) are *shaped rewards* that help individual agents filter out the *noise* from the global reward signal (which includes effects from other agents' actions), and assess their individual contribution to the global reward. As such, there is no general technique to compute DRs for different problems. We therefore develop novel methods to approximately compute two popular types of DRs—wonderful life utility (WLU) and aristocratic utility (AU) [41] for the collective case.

**Wonderful Life Utility (WLU):** Let $\boldsymbol{s}, \boldsymbol{a}$ denote the joint state-action; $r(\boldsymbol{s}, \boldsymbol{a})$ be the system reward. The WLU based DR for an agent $m$ is $r^m = r(\boldsymbol{s}, \boldsymbol{a}) - r(\boldsymbol{s}, \boldsymbol{a}^{-m})$ where $\boldsymbol{a}^{-m}$ is the joint-action without the agent $m$. The WLU DR compares the global reward to the reward received when agent $m$ is not in the system. Agent $m$ can use this shaped reward $r^m$ for its individual learning. However extracting such shaped rewards from the simulator is very challenging and not feasible for large number of agents. Therefore, we apply this reasoning to the critic (or action-value function approximator) $Q_w(\mathrm{n}^{sa}, d)$. Similar to WLU, we define WLQ (*wonderful life Q-function*) for an agent $m$ as $Q^m = Q_w(\mathrm{n}^{\mathrm{sa}}, d) - Q_w(\mathrm{n}^{\mathrm{sa}-m}, d)$ where $\mathrm{n}^{\mathrm{sa}-m}$ is the state-action count table without the agent $m$.

For a given $(\mathrm{n}^{\mathrm{sa}}, d)$, we show how to estimate $Q^m$. Assume that the agent $m$ is in some state $i \in S$ and performing action $j \in A$. As agents do not have identities, we use $Q^{ij}$ to denote the WLQ for any agent in state-action $(i, j)$. Let $e^{ij}$ be a vector with the same dimension as $\mathrm{n}^{\mathrm{sa}}$; all entries in $e^{ij}$ are zero except value 1 at the index corresponding to state-action $(i, j)$. We have $Q^{ij} = Q_w(\mathrm{n}^{\mathrm{sa}}, d) - Q_w(\mathrm{n}^{\mathrm{sa}} - e^{ij}, d)$. Typically, critic $Q_w$ is represented using a neural network; we normalize all count inputs to the network (denoted as $\tilde{\mathrm{n}}^{\mathrm{sa}} = \mathrm{n}^{\mathrm{sa}}/M$) using the total agent population $M$. We now estimate WLQ assuming that $M$ is large:

$$Q^{ij} \approx \lim_{M \to \infty} \left[ Q_w\left(\mathrm{n}^{\mathrm{sa}}/M, d\right) - Q_w\left((\mathrm{n}^{\mathrm{sa}} - e^{ij})/M, d\right) \right] = \lim_{\Delta = 1/M \to 0} \left[ Q_w\left(\tilde{\mathrm{n}}^{\mathrm{sa}}, d\right) - Q_w\left(\tilde{\mathrm{n}}^{\mathrm{sa}} - \Delta \cdot e^{ij}, d\right) \right]$$

$$= -1 \cdot \lim_{\Delta = 1/M \to 0} \left[ Q_w\left(\tilde{\mathrm{n}}^{\mathrm{sa}} - \Delta \cdot e^{ij}, d\right) - Q_w\left(\tilde{\mathrm{n}}^{\mathrm{sa}}, d\right) \right] \quad (4)$$

$$= -1 * (-\Delta) \frac{\partial Q_w}{\partial \tilde{\mathrm{n}}^{\mathrm{sa}}(i,j)}(\tilde{\mathrm{n}}^{\mathrm{sa}}, d) \qquad \text{(by definition of total differential)}$$

$$Q^{ij} \approx \frac{1}{M} \frac{\partial Q_w}{\partial \tilde{\mathrm{n}}^{\mathrm{sa}}(i,j)}(\tilde{\mathrm{n}}^{\mathrm{sa}}, d) \quad (5)$$

Thus, upon experiencing the tuple $(\mathrm{n}_t^{\mathrm{s}}, d_t, \mathrm{n}_t^{\mathrm{sa}}, \mathrm{n}_t^{\mathrm{sas}}, d_{t+1}, r_t)$, global reward $r_t$ is used to train the global critic $Q_w$. An agent $m$ in state-action $(i, j)$ accumulates the gradient term $Q^{ij} \nabla_\theta \log \pi_t(j|i, o(i, d_t, \mathrm{n}_t^{\mathrm{s}}))$ as per the standard policy gradient result [37](notice that policy $\pi$ is the same for all the agents). Given that there are $\mathrm{n}_t^{\mathrm{sa}}(i, j)$ agents performing action $j$ in state $i$, the total accumulated gradient based on WLQ updates (5) by all the agents for all time steps is given as:

$$\nabla_\theta^{wlq} J(\theta) = \sum_{t=1}^{H} \mathbb{E}_{d_t, \mathrm{n}_t^{\mathrm{sa}} | b_o, b_o^d} \left[ \sum_{i \in S, j \in A} \mathrm{n}_t^{\mathrm{sa}}(i, j) Q_t^{ij}(\mathrm{n}_t^{\mathrm{sa}}, d_t) \nabla_\theta \log \pi_t(j|i, o(i, d_t, \mathrm{n}_t^{\mathrm{s}})) \right] \tag{6}$$

We can estimate $\nabla_\theta^{wlq} J(\theta)$ by sampling counts and the state $d_t$ for all the time steps.

**Aristrocratic Utility (AU):** For a given joint state-action $(\boldsymbol{s}, \boldsymbol{a})$, the AU based DR for an agent $m$ is defined as $r^m = r(\boldsymbol{s}, \boldsymbol{a}) - \sum_{a^m} \pi^m(a^m | o^m(\boldsymbol{s})) r(\boldsymbol{s}, \boldsymbol{a}^{-m} \cup a^m)$ where $\boldsymbol{a}^{-m} \cup a^m$ is the joint-action where agent $m$'s action in $\boldsymbol{a}$ is replaced with $a^m$; $o^m$ is the observation of the agent; $\pi^m$ is the probability of action $a^m$. The AU marginalizes over all the actions of agent $m$ keeping other agents' actions fixed. We next define AU-based reasoning to the critic $Q_w$. For a given $(\mathrm{n}^{\mathrm{sa}}, d)$, we define $A^{ij}$ as the counterfactual advantage function for the agent in state $i$ and taking action $j$ as:

$$A^{ij} = Q_w(\mathrm{n}^{\mathrm{sa}}, d) - \sum_{j'} \pi(j'|i, o(i, d, \mathrm{n}^{\mathrm{s}})) Q_w(\mathrm{n}^{\mathrm{sa}} - e^{ij} + e^{ij'}, d) \tag{7}$$

where vectors $e^{ik}$ are defined as for WLQ. Such advantages have been used by [16]. However in our setting, computing them naively is prohibitively expensive as the number of agents is large (in thousands). Therefore, we use similar technique as for WLQ by normalizing counts, and computing differentials $\lim_{\Delta = 1/M \to 0} \left[ Q_w(\tilde{\mathrm{n}}^{\mathrm{sa}}, d) - Q_w(\tilde{\mathrm{n}}^{sa} + \Delta \cdot (e^{ij'} - e^{ij}), d) \right]$, final estimate is (proof in appendix):

$$A_t^{ij}(\mathrm{n}_t^{\mathrm{sa}}, d_t) \approx \frac{1}{M} \left[ \frac{\partial Q_w}{\partial \tilde{\mathrm{n}}^{\mathrm{sa}}(i, j)}(\mathrm{n}_t^{\mathrm{sa}}, d_t) - \sum_{j'} \pi(j'|i, o(i, d_t, \mathrm{n}_t^{\mathrm{s}})) \frac{\partial Q_w}{\partial \tilde{\mathrm{n}}^{\mathrm{sa}}(i, j')}(\mathrm{n}_t^{\mathrm{sa}}, d_t) \right] \tag{8}$$

Crucially, the above computation is independent of agent population $M$, and is thus highly scalable. Using the same reasoning as WLQ, the gradient $\nabla_\theta^{au}$ is exactly the same as (6) with $Q_t^{ij}$ replaced by advantages $A_t^{ij}$ in (8). Empirically, we observed that using advantages $A^{ij}$ resulted in better quality because the additional term $\sum_{j'}$ in $A^{ij}$ acts as a baseline and reduces variance.

## 4 Mean Collective Actor Critc—Credit assignment, low variance gradients

Notice that computing gradients $\nabla_\theta^{au}$, $\nabla_\theta^{wlq}$ for DRs requires taking expectation over state-action counts $\mathrm{n}^{\mathrm{sa}}$ (see (6)), which can have high variance. Furthermore, the DR approximation is accurate only when the agent population $M$ is large; for smaller populations we empirically observed a drop in the solution quality using DRs. We next show how to address these limitations by developing a new approach called mean collective actor critic (MCAC) which is robust across a range of population sizes, and empirically works better than DRs in several problems.

- We develop an alternative formulation of the policy gradient (3) that allows to analytically marginalize out state-action counts $\mathrm{n}_t^{\mathrm{sa}}$. By analytically computing the expectation over counts, variance in the gradient estimates can be reduced, as also shown for MDPs in [11, 5].

- We show that a factored critic structure is particularly suited for credit assignment, and also allows analytical gradient computation by using results from collective graphical models [22].

- However, factored critic is not effectively learnable with global rewards. Our key insight is that we *learn a global critic* which is not factorizable among agents. Instead of computing gradients from this critic, we estimate gradients from its first-order Taylor approximation, which fortunately is factored among agents, and fits well within our previous two results above.

**Variance reduction of gradient using expectation:** Before reformulating the gradient expression (3), we first define $P^\pi(\mathrm{n}_t^{\mathrm{sa}} | \mathrm{n}_t^{\mathrm{s}}, d_t)$ as the collective distribution of the action counts given the action probabilities $\pi$ and state counts:

$$P^\pi(\mathrm{n}_t^{\mathrm{sa}} | \mathrm{n}_t^{\mathrm{s}}, d_t) = \prod_{i \in S} \left[ \frac{\mathrm{n}_t^{\mathrm{s}}(i)!}{\prod_{j \in A} \mathrm{n}_t^{\mathrm{sa}}(i, j)!} \prod_{j \in A} \pi(j|i, o(i, d_t, \mathrm{n}_t^{\mathrm{s}}))^{\mathrm{n}_t^{\mathrm{sa}}(i, j)} \right] \tag{9}$$

The above is a multinomial distribution—for each state $i$, we perform $\mathrm{n}_t^{\mathrm{s}}(i)$ trials independently (one for each of $\mathrm{n}_t^{\mathrm{s}}(i)$ agents). Each trial's outcome is an action $j \in A$ with probability $\pi(j|i, o(i, d_t, \mathrm{n}_t^{\mathrm{s}}))$.

**Proposition 1.** *The collective policy gradient in* (3) *can be reformulated as:*

$$\nabla_\theta J(\theta) = \sum_{t=1}^{H} \mathbb{E}_{\mathrm{n}_t^{\mathrm{s}}, d_t | b_o, b_o^d} \Big[ \sum_{\mathrm{n}^{\mathrm{sa}}} Q_t^\pi(\mathrm{n}^{\mathrm{sa}}, d_t) \nabla_\theta P^\pi(\mathrm{n}^{\mathrm{sa}} | \mathrm{n}_t^{\mathrm{s}}, d_t) \Big] \qquad (10)$$

Proof is provided in appendix. In the above expression we sample $(\mathrm{n}_t^{\mathrm{s}}, d_t)$ and analytically marginalize out state-action counts $\mathrm{n}_t^{\mathrm{sa}}$, which will result in lower variance than using (3) directly to estimate gradients. In the AC approach, we use a critic to approximate $Q^\pi$. However, not all types of critics will enable analytical marginalization over state-action counts.

**Critic design for multiagent credit assignment:** We now investigate the special structure required for the critic $Q_w$ that enables the analytical computation required in (10), and also helps in the multiagent credit assignment. One solution studied in several previous works is a linear decomposition of the critic among agents [36, 18, 20]: $Q_w(\boldsymbol{s}_t, d_t, \boldsymbol{a}_t) = \sum_{m=1}^{M} f_w^m(\boldsymbol{s}_t^m, \boldsymbol{a}_t^m, o(\boldsymbol{s}_t^m, d_t, \mathrm{n}_t^{\mathrm{s}}))$.

Such a factored critic structure is particularly suited for credit assignment as we are explicitly learning $f_w^m$ as an agent $m$'s contribution to the global critic value. Crucially, we also show that the policy gradient computed using such a critic also gets factored among agents, which is essentially credit assignment at the level of gradients among agents. In the collective setting, counts are the sufficient statistic for planning, and we assume a homogenous stochastic policy. Therefore, the critic simplifies as: $Q_w(\mathrm{n}_t^{\mathrm{sa}}, d_t) = \sum_{i,j} \mathrm{n}_t^{\mathrm{sa}}(i,j) f_w(i, j, o(i, d_t, \mathrm{n}_t^{\mathrm{s}}))$. The next result uses a more general definition of $f_w$ that may depend on entire state counts $\mathrm{n}_t^{\mathrm{s}}$. Proof (in appendix) uses results from Gaussian approximation of collective graphical models [22].

**Theorem 2.** *A linear critic,* $Q_w(\mathrm{n}_t^{\mathrm{sa}}, d_t) = \sum_{i,j} \mathrm{n}_t^{\mathrm{sa}}(i,j) f_w(i, j, d_t, \mathrm{n}_t^{\mathrm{s}}) + b(d_t, \mathrm{n}_t^{\mathrm{s}})$ *where function* $b$ *only depends on* $(d_t, \mathrm{n}_t^{\mathrm{s}})$, *has the expected policy gradient under the policy* $\pi^\theta$ *as:*

$$\sum_{\mathrm{n}^{\mathrm{sa}}} Q_w(\mathrm{n}^{\mathrm{sa}}, d_t) \nabla_\theta P^\pi(\mathrm{n}^{\mathrm{sa}} | \mathrm{n}_t^{\mathrm{s}}, d_t) = \sum_{i \in S, j \in A} \mathrm{n}_t^{\mathrm{s}}(i) \nabla_\theta \pi_t^\theta(j|i, o(i, d_t, \mathrm{n}_t^{\mathrm{s}})) f_w(i, j, d_t, \mathrm{n}_t^{\mathrm{s}}) \qquad (11)$$

**Learning the critic from global rewards:** The factored critic used in theorem 2 has two major disadvantages. First, learning the factored critic from global returns is not effective as crediting empirical returns into contributions from different agents is difficult to learn without requiring too many samples. Second, the critic components $f_w$ are based on an agent's local state, action while ignoring other agents' policy and actions which increases the inaccuracy as both local and global rewards are affected by other agents' actions.

Our key insight is that instead of learning a decomposable critic, we *learn a global critic* which is not factorized among agents. This addresses the problem of learning from global rewards; as the critic is defined over the input from all the agents (count tables **n** in our case). However, instead of computing policy gradients directly from the global critic, we compute gradients from a *linear approximation* to the global critic using first-order Taylor approximation. Actor update using linear approximation of the critic is studied previously for MDPs in [11, 32]. Given a small step size, the linear approximation is sufficient to estimate the direction of the policy gradient to move towards a higher value. For our case, linear critic addresses both the credit assignment problem and low variance gradient estimates. Consider the global critic $Q_w(\mathrm{n}_t^{sa}, d_t)$, we consider its first order Taylor expansion at the mean value of action counts $\mathrm{n}_t^{\star\,\mathrm{sa}} = \mathbb{E}[\mathrm{n}_t^{\mathrm{sa}} | \mathrm{n}_t^{\mathrm{s}}, d_t] = \langle \mathrm{n}_t^{\mathrm{s}}(i) \pi(j|i, o(i, d_t, \mathrm{n}_t^{\mathrm{s}})) \forall i, j \rangle$ with $\pi$ as the current policy:

$$Q_w(\mathrm{n}_t^{\mathrm{sa}}, d_t) \approx Q_w(\mathrm{n}_t^{\star\,\mathrm{sa}}, d_t) + (\mathrm{n}_t^{\mathrm{sa}} - \mathrm{n}_t^{\star\,\mathrm{sa}})^\intercal (\nabla_{\mathrm{n}^{\mathrm{sa}}} Q_w(\mathrm{n}^{\mathrm{sa}}, d_t)|_{\mathrm{n}^{\mathrm{sa}} = \mathrm{n}_t^{\star\,\mathrm{sa}}}) \qquad (12)$$

Upon re-arranging the above, it fits the critic structure in theorem 2:

$$Q_w(\mathrm{n}_t^{\mathrm{sa}}, d_t) \approx \sum_{i,j} \mathrm{n}_t^{\mathrm{sa}}(i,j) \frac{\partial Q_w}{\partial \mathrm{n}^{\mathrm{sa}}(i,j)}(\mathrm{n}_t^{\star\,\mathrm{sa}}, d_t) + \Big[ Q_w(\mathrm{n}_t^{\star\,\mathrm{sa}}, d_t) - (\mathrm{n}_t^{\star\,\mathrm{sa}})^\intercal (\nabla_{\mathrm{n}^{\mathrm{sa}}} Q_w(\mathrm{n}^{\mathrm{sa}}, d_t)|_{\mathrm{n}^{\mathrm{sa}} = \mathrm{n}_t^{\star\,\mathrm{sa}}}) \Big]$$

Using theorem 2 and proposition 1, we have (proof in appendix):

**Corollary 1.** *Using the first-order Taylor approximation of the critic at the expected state-action counts* $\mathrm{n}_t^{\star\,\mathrm{sa}} = \mathbb{E}[\mathrm{n}_t^{\mathrm{sa}} | \mathrm{n}_t^{\mathrm{s}}, d_t; \pi]$, *the collective policy gradient is:*

$$\nabla_\theta J(\theta) \approx \sum_{t=1}^{H} \mathbb{E}_{\mathrm{n}_t^{\mathrm{s}}, d_t | b_o, b_o^d} \Big[ \sum_{i \in S, j \in A} \mathrm{n}_t^{\mathrm{s}}(i) \nabla_\theta \pi_t(j|i, o_t(i, d_t, \mathrm{n}_t^{\mathrm{s}})) \frac{\partial Q_w}{\partial \mathrm{n}^{\mathrm{sa}}(i,j)}(\mathrm{n}_t^{\star\,\mathrm{sa}}, d_t) \Big] \qquad (13)$$

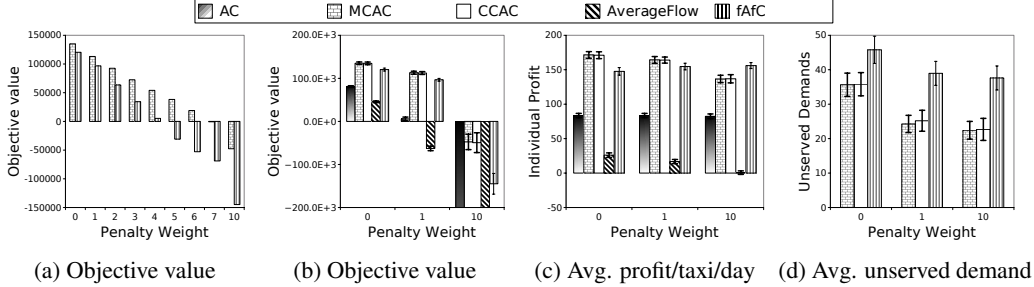

Figure 2: Different metrics on the taxi problem with different penalty weights $w$.

Intuitively, terms $\partial Q_w / \partial n(i,j)$ facilitate credit assignment, which also occur in DR based formulations (section 3). When this term has a high value, it implies that a higher count of agents in state $i$ and taking action $j$ would increase the overall critic value $Q$. This will encourage more agents to take action $j$ in state $i$. Each term $\partial Q_w / \partial n(i,j)$ is evaluated at the overall state-action counts $n_t^{\star \, \text{sa}}$ which in turn depend on the policy and actions of other agents. Thus, it overcomes the second limitation of the factored critic in theorem 2 where terms $f_w$ ignore policy and actions of other agents.

## 5  Experiments

We test the aristocratic utility based approach (called 'CCAC' or collective counterfactual AC) that uses gradient estimates (8), and the mean collective AC ('MCAC') that uses (13). We test against (a) the standard AC approach which fits the critic using global rewards and computes gradients from the global critic; (b) the factored actor critic ('fAfC') approach of [26], the previous best approach for $\mathbb{C}$Dec-POMDPs with *decomposable* rewards; (c) the average flow based solver ('AverageFlow') of [42]. In some domains (specifically the taxi problem), we have both local and global rewards. The local rewards are incorporated in 'fAfC' as before; for global rewards, we change the training procedure of the critic in 'fAfC' (different AC updates are shown in appendix). We test on two real world domains—taxi supply-demand matching [43], and the police patrolling problem [10].

**Taxi Supply-Demand Matching:** The dataset consists of taxi demands (GPS traces of the taxi movement and their hired/unhired status) in an Asian city over 1 year. The fleet contains 8000 taxis (or agents) with the city divided in 81 zones. Environment dynamics are similar to [43]. The environment is uncertain (due to stochastic demand), and partially observable as each taxi observes the count of other taxis and the demand in the current zone and geographically connected neighbor zones, and decides its next action (stay or move to a neighboring zone). Over the plan horizon of 48 half hour intervals, the goal is to compute policies that enable strategic movement of taxis to optimize the total fleet profit. Individual rewards model the revenue each taxi receives. *Global rewards* model *quality-of-service* (QoS) by giving a high positive reward when the ratio of available taxis and the current demand in a zone is greater than some threshold, and negative reward when the ratio is below the set QoS. We selected the topmost 15 busiest zones for such global rewards. To enforce QoS level $\alpha = 95\%$ for each zone $i$ and time $t$, we add penalty terms $\min(0, w \times (\hat{d}_t(i) - \alpha d_t(i)))$ where $w$ is the penalty weight, $\hat{d}_t(i)$ is the total *served* demand at time $t$, and $d_t(i)$ is the total demand at time $t$. We test the effect of QoS penalty by using weights $w \in [0, 10.0]$. We normalize all trip payments between $(0, 1)$ which implies that the penalty for missing a customer over the QoS threshold is roughly $w$ times the negative of the maximum reward for serving a customer.

Figure 2(a) shows the quality comparisons (higher is better) between MCAC (CCAC is almost identical to MCAC) and fAfC with varying penalty $w$. It shows that with increasing $w$, fAfC becomes significantly worse than MCAC. We next investigate the reason. Figure 2(b) summarizes quality comparisons among all approaches for three settings of $w$. Results confirm that both MCAC and CCAC provide similar quality, and are the best performing among the rest. 'AverageFlow' and 'AC' are much worse off due to presence of global rewards. As the weight $w$ increases from 0 to 10, the difference between CCAC/MCAC and fAfC increases significantly. This is because higher $w$ puts more emphasis on optimizing global rewards. Figure 2(d) shows unserved demand below the QoS threshold or $(\alpha \cdot d_t(i) - \hat{d}_t(i))$ averaged over all 15 zones and all the time steps (AC, AverageFlow are omitted as their high numbers distort the figure). When penalty increased from $w = 0$ to $1$ in

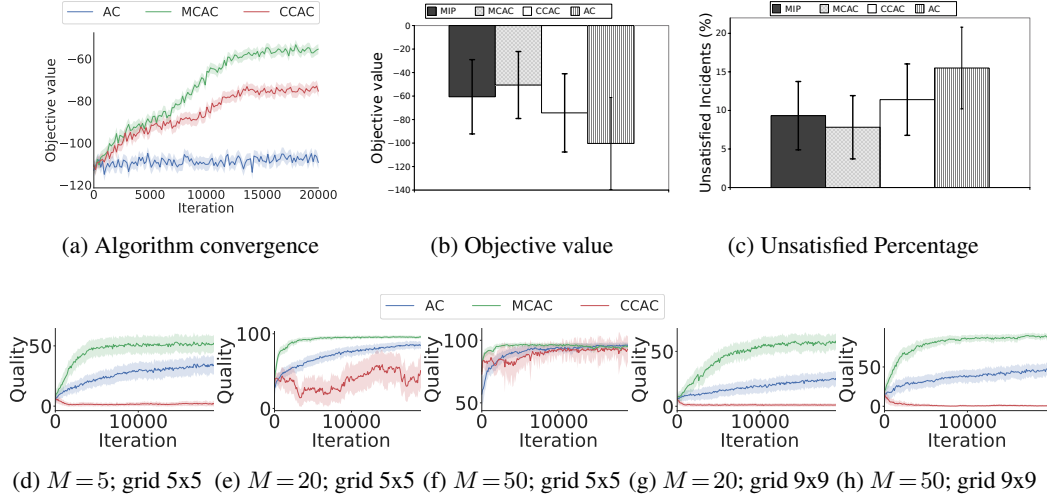

(a) Algorithm convergence      (b) Objective value      (c) Unsatisfied Percentage

(d) $M=5$; grid 5x5   (e) $M=20$; grid 5x5   (f) $M=50$; grid 5x5   (g) $M=20$; grid 9x9 (h) $M=50$; grid 9x9

Figure 3: (a)-(c) Police patrolling problem; (d)-(h) synthetic grid patrolling with varying population $M$, grids

figure 2(c), MCAC/CCAC still maintain similar individual profits, but their unserved demand decreased significantly (by 32%) as shown in figure 2(d). Thus, CCAC/MCAC maintain individual profits while still reducing global penalty, and are therefore effective with global rewards. In contrast, the unserved demand by fAfC does not decrease much from $w=0$ to $w=1, 10$; because the QoS penalty constitutes global rewards whereas 'fAfC' is optimized for decomposable rewards.

**Police Patrolling:** The problem is introduced in section 2. There are 24 city zones, and 16 patrol cars (or agents). We have access to real world data about all incidents for 31 days in 24 zones. Roughly 50-60 incidents happen per day (7AM-7PM shift). The goal is to compute reallocation policy for agents such that number of incidents with response time more than the threshold is minimized (further details in appendix). This domain has only global rewards. Therefore, we compare MCAC, CCAC and AC (fAfC, AverageFlow are unable to model this domain). As a baseline, we compare against a static allocation of patrol cars that is optimized using a stochastic math program [10], denoted as 'MIP'. Figure 3(a) shows the convergence results. MCAC performs much better than CCAC. This is because this problem is sparse with sparse tables $n^{sa}$, resulting in higher gradient variance for CCAC; MCAC marginalizes out $n^{sa}$, thus has lower variance. Figure 3(b) shows overall objective comparisons (higher is better) among all three approaches. It confirms that MCAC is the best approach. MCAC has 7.8% incidents where response time was more than the threshold versus 9.32% for MIP (figure 3(c)). Notice that even this improvement is significant as it allows $\approx$25 more incidents to be served within the threshold over 31 days (assuming 55 avg. incidents/day). In emergency scenarios, improving response time even by few minutes is potentially life saving.

To further compare CCAC and MCAC, we created a *synthetic grid patrolling* problem also inspired by police patrolling, where we vary grid sizes and agent population (domain details in appendix). Figure 3(d-h) show convergence plots. In these problems, CCAC performs much worse (even worse than AC) as these problems are sparse with sparse state-action counts $n^{sa}$. This makes its gradient variance higher than MCAC, which again performs best. To summarize, when the population size is large and state-action counts are dense (as in the taxi problem with $M=8000$), both CCAC and MCAC give similar quality; but for small population size (as in grid patrolling with $M=5$), MCAC is more robust than CCAC and AC.

## 6 Summary

We developed several approaches for credit assignment in collective multiagent RL. We extended the notion of difference rewards to the collective setting and showed how to compute them efficiently even for very large agent population. To further reduce the gradient variance, we developed a number of results that analytically marginalize out agents' actions from the gradient expression. This approach, called MCAC, was more robust than difference rewards based approach across a number of problems settings and consistently provided better quality over varying agent population sizes.

# 7 Acknowledgments

This research project is supported by National Research Foundation Singapore under its Corp Lab @ University scheme and Fujitsu Limited. First author is also supported by A*STAR graduate scholarship.

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
