[Reviews · NeurIPS 2018]

Reviewer 1



The paper tackles a multi-agent credit assignment problem, an egregious problem within multi-agent systems by extending existing methods on difference rewards for settings in which the population of the system is large. Though the results are relevant and lead to an improvement for large population systems, the contribution is nonetheless limited to a modification of existing techniques for a specific setting which seemingly requires the number of agents to be large and for the agents to observe a count of the agents within their neighbourhood. The results of the paper enable improved credit assignment in the presence of noise from other agents' actions, an improved baseline leading to reduced variance and, in turn, better estimates of the collective policy gradient (under homogeneity assumptions). The analysis of the paper applies to a specific setting in which the reward function has a term that is common to all agents and therefore is not decomposable. The extent to which this property is to be found in multi-agent systems, however, is not discussed. The setting is also partially observable multi-agent system - an instance of which is motivated by the authors. This, as the authors claim, implies that the agents need not observe the full state but need only make local observations for which the problem is formulated using a decentralised POMDP framework. The claims of the paper are supported by some theoretical results though some important details on which the theory rely seem to be omitted. Additionally, the analysis of the paper is accompanied by some experiments that showcase the benefits of the approach. A weakness of the general method is that the approach seems to alleviate computational complexity during execution of a learned policy but during the learning phase, however, the method seems not to alleviate the problem of explosive complexity in the number of agents which may cause tractability issues for large populations. Furthermore, the method seems reliant on the population size being large for the approximations to hold though the paper does not include a discussion on when the population size is reasonably large for the method to be valid. Finally, though the contribution is useful for the given setting, it is not clear how often the central assumption of non-decomposability occurs in multi-agent systems. (edited) There are also some other important questions that if addressed would improve the paper: In particular, though the setting of the paper is said to be a partially observable - it seems that 'distant' agents do not have an immediate effect on the agents’ rewards since the rewards depend on variables that are defined locally (though they may interact with the agent at some later time). In this sense each agent's observations fully capture all relevant state information - it doesn't seem as though extra information concerning the agent's immediate reward or state transition can be inferred by including global information. Secondly, the authors introduce the notion of agent types as a way of introducing heterogeneity into the formalism. The notion of agent types, however, seems inconsistent since it is assumed that the agents have the same policy which would differ for different agent types if type is any meaningful parameter (i.e. determines the agents' rewards/state transition probabilities). Additionally, heterogeneity from agent types also seems not to fit with the idea of a global critic. Lastly, the advantage function (15) as used in the cited paper is used in a team game setting when agents have the same reward function. There are a number of details on which the theory is built that are omitted - in particular: 1. The authors do not state conditions under which sufficient statistic is actually sufficient (is this derived from application of the law of large numbers?) - this also seems to impose conditions of homogeneity on the agents - without a clear statement of the conditions under which this holds, some other statements are hard to evaluate e.g. line 163: "which makes learning scalable even with very large number of agents" and "Crucially, the above computation is independent of agent population M"- it seems that these statements in fact only hold in cases when M is large. This also arises in lines 195-196 where M must be large for the approximation to be valid as well as in the proof B1. 2. The authors state that "Previous work shows that count tables are the sufficient statistic for planning" - this seems to be fairly fundamental, it would be of great benefit to the reader to have a citation and a reference to the result for this remark. The paper would also benefit from a discussion on the solution admitted by the method. Specifically, though the setting is non-cooperative since each agent maximises their individual rewards the examples discussed nonetheless suggest the authors seek to target cooperative or team games. Though cooperative and team-based games can be tackled in a decentralised way, the stable points in non-cooperative game setting (i.e. Markov game) are described by Nash equilibria (Markov-perfect Nash equilibria) which are in general inefficient from a social welfare perspective. The authors ought to include a discussion on how they expect the payoffs from the learned policies which seem to be generated by Nash equilibrium strategies to perform in relation to the system goal in general and, possibly how they might overcome challenges of inefficiency from non-cooperative challenges. 3. What is the meaning of d - can the authors give an example? 4. Additionally, the authors state that "We assume that Qw is differentiable in all input parameters". This doesn't seem to be a reasonable assumption given that one of the inputs n^{sa} is a discrete integer count. The reason for my score is the following. The paper is clearly written leading to a coherent exposition and its objectives are clearly stated. The contribution seems however only incremental given the existing (and cited) methods on difference rewards and in the absence of clear statements when the method is most applicable. The crucial element, however, is about the correctness of the method. I have two major concerns: i) about the differentiability of Q_w* and ii) about small population sizes. There's also a (minor) issue about the claim that the method can work with heterogeneous populations, where the method, which seems to depend on one single policy being learned, may break down. i) The assumption of differentiability of Q_w*, which appears to be (likely) invalidated in the current setting as one of its inputs is a discrete integer variable. This seems to me to be a concern as differentiation here is ill-defined. The authors have not addressed this issue in their rebuttal. ii) The derivations of the Q function estimates rely on large populations (e.g. eqs. (5), (9)), and use previous results (e.g. citations [4] and [6] on the appendix) but those papers (beside having many notational inconsistencies) also rely on large populations. Overall, the paper is well written and easy to follow, but I have major concerns with the correctness of the method. Also, reading the previous papers that this paper heavily relies on, the contribution looks very incremental.

Reviewer 2



Overall this is an interesting, well written article that tackles multi-agent credit assignment in the context of complex multi-agent learning, where the reward structure is not decomposable amongst agents. They experiment in several settings, including taxi fleet optimization and multiagent patrolling. Overall, I think this is a well written, well structured piece of research that presents some new contributions building on earlier work in the area, such as difference rewards, but now making these ideas applicable when there is a global reward that is no longer decomposable, integrated with an actor-critic MAL gradient approach. It is nice to see that these ideas can now be applied in real-world settings, and I appreciate the fact that the models are based on real-world data. The authors propose two new algorithms (CCAC and MCAC) that help deal with credit assignment in a multi-agent RL setting that seem to outperform previous algorithms on benchmarks used in literature before. But then the domains used also have applicability in the real world - a police patrol domain and a taxi supply domain.  
Rigorous mathematical proofs backing up the claims they make are provided in the supplementary material and the algorithmic construction seem to come out of sound theoretical ideas.

Reviewer 3



In this paper, the authors propose a new algorithm, inspired by actor-critic methods, for multi-agents setting in partially observable environments under a collective global reward objective function. Moreover, the authors provide a credit assignment method in order to boost up the learning. In this work, the agents are considered to be equal (exchangeable), therefore follow the same policy. The paper is well written and has a clear overview of the previous works. Comments: If the proposed method is for just planing and not for RL, I would suggest changing the title, the proposition, and also the motivation. It might mislead the readers. Moreover, comparison to Monte Carlo search is also motivated. The authors start to discuss the decentralized learning, but the method developed seems to be centralized since a single Q function is learned. Also, it would be better to clarify whether the agents observe the latent state "d". If yes, please elaborate on Dec-POMDP discussion, if not, please elaborate more on how the Q function is learned in a decentralized fashion. Also does each agent observes others action? Moreover is the policy shared? If no, how the updates occur and if yes, is it really a multi-agent setting? Generally, the optimization here does not happen in the decentralized fashion (at least I could not see the decentralized part), therefore please provide a more clear motivation for your solution and clarify why is not solving a single MDP. Also, more motivation over two provided utility functions is useful. Also, Apex F is missing.